# The Effect of Virtual-Reality-Based Restorative Environments on Creativity

**DOI:** 10.3390/ijerph191912083

**Published:** 2022-09-24

**Authors:** Hongqidi Li, Xueyan Du, Huirui Ma, Zhimeng Wang, Yue Li, Jianping Wu

**Affiliations:** 1Department of Psychology, School of Humanities and Social Sciences, Beijing Forestry University, Beijing 100083, China; 2Faculty of Psychology, Beijing Normal University, Beijing 100875, China; 3Department of Psychology, Lund University, 221 00 Lund, Sweden

**Keywords:** restorative environment, virtual reality, creativity, presence, EEG

## Abstract

This study, based on the theory of restorative environmental, uses virtual reality (VR) technology to construct interactive restorative environments and discusses the influence of the experience of virtual restorative environment on individual creativity. A total of 72 college students were selected as participants in the study. Through psychological scales, three creativity tests, and EEG feedback data, the following conclusions were drawn: (1) The VR restorative environment experience improves individual creativity, especially the creative quality of cohesion; (2) the experience of the VR restorative environment enables participants to experience a desirable sense of presence. Compared with the restorative scene experience without interactive activities, the addition of interactive activities improves the individual sensory fidelity to a greater extent. (3) We cannot simply assume that the experience of the VR restorative environment with interactive activities will make individual creative performance better than non-interactive experience. Interaction with certain difficulty will increase cognitive load, thus disrupting individual creative performance. Garden scenes that can be explored freely and have no interaction can better promote individual creativity. (4) In the environmental experience, participants paid greater attention to natural elements, and the restorative environment they described was very similar to the environment they believed could foster creativity. This study’s results provide evidence for the positive effects of the VR restorative environment experience on individuals and contributes to the cognitive exploration of the interaction between restorative environments and individuals in the future.

## 1. Introduction

Creativity is the intellectual ability of using one’s entire knowledge base to produce something new, unique, and socially or personally valuable toward a specific purpose. Most scholars agree that creativity is related to novel products, ideas, or valuable problem-solving methods [1]. This has been a contentious research topic in philosophy, physiology, and neuroscience, as well as psychology. Creativity psychologically benefits individuals. Creativity fulfills people’s unique needs and improves their mental health by helping them develop self-efficacy, realize their self-worth, and experience the joy of self-actualization through creative activities [2,3]. Studies have found that creativity is positively correlated with positive emotions and happiness [4,5,6,7,8]. Additionally, creativity is an effective resource for individuals in crisis [9,10]. Engaging in creative activities in response to difficult events or during times of crisis (such as the COVID-19 pandemic) can promote wellbeing and help individuals in collectivist countries (such as China) experience greater social connection, which helps them cope better with social isolation and loneliness caused by social distancing during the epidemic [11]. Therefore, the study of creativity is of great significance to the positive development of individuals and society.

Research on the factors influencing creativity has provided a theoretical basis for its nurturing strategies and developmental pathways. Social environmental factors, such as family and school; physical environmental factors, such as brightness; and cognitive factors, such as intelligence, cognitive style, and knowledge level; as well as personality and motivation, all exhibit a significant impact on creativity [12,13,14,15]. Although human creativity is a relatively stable trait, any external factors that stimulate the creator’s thinking experience can also influence creativity, with the environment being an important example [16]. A desirable environment facilitates—whereas an undesirable environment hinders—the formation and development of creativity [17,18].

Researchers have empirically examined the relationship between social and physical environments and creativity, and have demonstrated that physical environmental stimulation affects an individual’s creative performance. Contact with natural environments, which has become a positive strategy for dealing with nature-deficit disorder in children, enhances their sensory abilities, facilitates improved physical and social skills, and stimulates their imagination and creativity [19]. The learning environment significantly contributes in supporting students’ creativity [20]. Studies have shown that environmental experiences may interfere with the impact of design major students’ cognitive styles on creativity [21]. Interior space design and environmental design elements—such as low complexity, more plants, adequate lighting, windows, cool tones—stimulate the creativity of workers in the workplace [22,23,24,25,26]. The basic properties of the physical work environment (PWE) influence creativity, with round-PWE being more likely to enhance divergent thinking and angular-PWE more likely to enhance convergent thinking [27]. Future research should further clarify which environmental settings positively affect individual creativity and how they affect the temporal dynamics of creativity promotion [28].

A growing body of recent research has shown that certain types of environmental stimuli, such as stimuli from the natural environment, enhance human creativity. Natural environments are full of interesting stimuli, which attract attention moderately from the bottom up such that top-down-directed attention has the opportunity to be supplemented; on the contrary, urban environment stimulation attracts significant attention, especially directed attention, and, thus, its restorative power is lower [29]. Nature helps supplement the directed attention required to generate new ideas by making people curious, evoking creative thinking in a more flexible way [30]. Working in an environment that contains natural elements is more conducive to the creativity of design professionals [31]. Studies have shown that wilderness hiking and forest healing have increased participants’ performance on creative tasks to varying degrees [32,33,34], and that mental processes that occur in natural environments, such as wandering, trigger greater flexibility in thinking, thus generating more new ideas [35]. Therefore, such restorative environments, whether artificial or natural, significantly impact creative performance. These studies have demonstrated that restorative environments, which are characterized by being away, extent, fascination, and compatibility [36,37], contribute in enhancing individual creativity. The theoretical underpinnings of research on restorative environments include attention restoration theory (ART) and stress reduction theory (SRT). ART states [38] that tasks requiring mental effort evoke directed attention, and that if the task is of a higher duration and intensity, even if the goal is pleasurable, it can cause mental exhaustion. Natural scenes provide stimuli that activate undirected attention and restore the depleted attentional system. SRT [37] explains the restorative effects of the environment from an emotional perspective, suggesting that individuals in stressful situations are drawn to environments of moderate depth and complexity, with visual focal points and containing vegetation and bodies of water. These environments aid in blocking negative thoughts, turning emotions positive, and restoring balance in the case of physiological disturbances, thus restoring healthy cognition and behavior.

Natural environments not only help supplement the directed attention needed to generate new ideas but also significantly contribute in stimulating creative thinking by allowing for greater curiosity, access to new ideas, and a more flexible way of thinking, especially in the first two stages of creativity—the preparation and incubation phases [30]. Activities in a restorative environment, such as field trips and forest healing, facilitate improved performances on creativity tests [32,33,39]. Designers who work in an indoor work environment containing natural elements exhibit higher levels of creativity in the product solutions that they develop compared to those generated in other environments, suggesting that such restorative environments, whether real or artificial, are conducive to creative product creation [31,40]. There is growing evidence that restorative environmental experiences enhance creativity. Williams assessed the mechanisms of restorative environments’ influence, proposing that a shift occurs between fascination with the external environment and internal wandering during natural experiences, that this shift provides the basis for attentional control pathways that enhance natural experiences, and that the complementarity of attentional recovery and wandering provides creativity enhancement benefits and generates new ideas [35].

Currently, studies on restorative environments have focused largely on the effects of experiences in natural environments—such as forests, nature reserves, wetlands, and urban parks—on mood, stress recovery, and mental health. Regarding the use of virtual reality (VR) technology in clinical research, it has been actively applied to the study of restorative environments from a health perspective, including the exploration of higher mental activities such as attention and cognition. Restorative environmental experiences have been shown to influence cognition and, to some extent, stimulate individual creativity, thereby leading to improved mental states and job performance [34,41]. Virtual nature may be a useful complement to actual nature [42], with more attractive and coherent features of a restorative environment [43]. VR natural environments relieve stress and help improve mood [44]. In comparison studies with virtual urban environments, virtual nature experiences exhibited greater positive psychological effects [45]. A 10 min virtual restorative experience positively impacts mood and self-efficacy, as reflected in prefrontal EEG indicators [46]. These recovery effects have been demonstrated in realistic natural scenarios [47,48], 2D pictures and videos [49], and immersive VR [50]. Compared to studies in real environments, which are limited to a single type of natural environment with limited monitoring of physiological indicators, VR technology permits the transition to laboratory settings, thus facilitating the collection of physiological data for the study of restorative environments. The simulated environment, unlike real environments, allows for the control of factors such as weather and lighting, thus enabling the randomized sequential presentation of multiple scenes in a short period of time, overcoming time and space constraints, and facilitating multisensory experiments and physiological data collection [51]. BBR, EI, and TBR are commonly used in the field of human–computer interaction and cognitive psychology, which can reflect the brain activity sensitivity of VR experiencers [52]. Through VR technology, simulating a perceptual experience consistent with—or even beyond—the reality of a natural environment triggers a psychological experience as if actually in such an environment.

Interactive forms of VR can exhibit a positive effect by attracting more attention and encouraging the experiencer to be in wider contact with the natural elements of the scene, creating a sense of “virtual presence”. Presence is a feeling of being present in a mediated environment, a mental state of “being there” [53]. Presence is regulated by environments that attract the senses and attention, and that promote active participation [54]. Schubert stated that Glenberg’s framework of embodied cognition explains the sense of presence, in that the direction and steps through which cognitive processes occur are determined by the body’s physical properties, wherein cognitive content is provided by the body and the body is embedded in the environment [55,56]. Thus, cognition, body, and environment form a dynamic unity, and cognition is intrinsically linked to embodiment and activity schemas [56]. The process behind being present is consistent with the theory of embodied cognition, which states that cognition is generated through physical experience and its activity patterns [57]. The activities in the virtual reality scene increase the physical participation of the participants, so that they have a better sense of presence and a more immersive feeling in the experience process, which better simulates the participants’ reaction in the real scene. Based on previous studies, the immersion and interactivity characteristics of virtual reality are closely related to creativity performance [58], and people are more likely to show creativity during interactive processes or highly immersive interactions. When activities stimulate visual interest, excitement, and freshness for entertainment, intellectual, and cognitive stimulation, they also promote innovative behaviors that help to improve professional knowledge and creative skills, stimulate motivation, and enhance creativity. When individuals participate in a suitable environment, even a non-real environment, such as a simulated natural environment, may affect behavior and perception, and stimulate creativity in a certain aspect [59].

The current research status of the restorative environment mainly includes the following points: first, the current research on the restorative environment is more based on the real environment, and the environment type is limited to a single natural environment. Second, the focus of the research has gradually shifted from individual mood and stress improvement to higher cognitive processes such as attention recovery and creativity improvement. Third, in the current study, physiological indicators in participants’ experience are less monitored. Considering that future research is likely to extend to different types of restorative environments, the trend is shifting toward laboratory experiments, and the advantages of VR technology to facilitate laboratory research and physiological data collection are evident. This study proposes to investigate the effects of VR-based restorative environment experiences on presence and individual creativity using both questionnaires and EEG readings, with college students as participants. It contributes to an improved understanding of the mechanisms underlying the restorative effects of natural environments, complements research on the effects of such environmental experiences on creativity, and provides a theoretical basis for the practical application of environmental restoratives.

Following existing research, it is reasonable to assume that the restorative environmental experience based mainly on a natural environment, and the experience based on differences in the sense of presence, enhance and impact the creative performance of college students in different ways. The specific assumptions are as follows.

**Hypothesis** **1.**
*The VR restorative environment experience enhances individual creativity, and such an experience provides participants a desirable sense of presence.*


**Hypothesis** **2.**
*The inclusion of interactive activities in VR restorative environment experiences significantly improves an individual’s sense of presence and creativity performance, outperforming non-interactive VR restorative environment experiences, as reflected in EEG metrics.*


**Hypothesis** **3.**
*Some features of the restorative environment are similar to those of an environment that can stimulate creativity, that is, the restorative environment may be a kind of environment that can stimulate individual creativity.*


## 2. Method

### 2.1. Participants

Participants were recruited through posters and informed of possible risks and privacy protection regulations. A total of 72 full-time undergraduate and graduate students with normal or corrected vision and no color blindness were selected as participants. There were 20 males and 52 females. They ranged in age from 18 to 29, with an average age of 21.58. The participants came from law, Japanese, e-commerce, information science, forestry, landscape architecture, mechanical design, manufacturing and automation, biotechnology, and other majors.

### 2.2. Materials and Instruments

Three VR scenes were constructed using the Unity interactive development system—urban scene, garden scene (a visual experience, including four areas: lawn, garden, water feature, and forest), and interactive garden scene (an interactive experience; four interactive activities were added to the garden scene, including kite flying, fishing, bird feeding, and watering crops) [46].

Creativity test:

(1) Adopting the Guilford multipurpose task used in Simone’s study to assess participants’ creativity, the Alternative Uses Task (AUT) score comprises four dimensions, namely, fluency, flexibility, originality, and persistence [60,61]. The fluency score was the sum of the number of valid answers given by the participants, and each score was one point. The flexibility score was the sum of the number of valid answer types given by the participants, and each type was scored as one point. The originality score was calculated using a subjective scale [62], in which all participants’ answers were combined to form a pool of answers, and each answer was scored by three examiners on a 5-point scale, with 1 being uncreative and 5 being very creative. The originality score was the average of the scores that their answers received. Persistence was the number of uses proposed for the item by the participants for the dimensions of fluency and flexibility. The total creativity score was the sum of the scores of the four dimensions.

(2) The Test for Creative Thinking, Drawing Production (TCT-DP) [63] is a holistic, nonverbal measure of creative ability. The task comprises five graphic elements within a rectangular frame and one element outside the frame. Two different versions (including a 180-degree rotated version) were used. Participants were asked to draw the drawing using the elements or pieces in any way they chose. The scoring criteria included 14 dimensions, with a maximum score of 6 points for each dimension and a maximum total score of 84 points.

(3) Chinese Version of the Compound Remote Associates Test (CRAT) [64]: Each question contains three Chinese characters, and only one character can be combined with these three characters to form words. A total of 100 questions were selected as experimental material, with one point for each correct answer.

The Restoration Environment Scale (RES), used to assess individuals’ perceived restoration, was developed by Liu, based on ART theory [65]. It is the first Chinese version of RES and includes the following dimensions: distance, attraction and compatibility, and enrichment. The Cronbach’s alpha coefficients of the total scale and three subscales ranged from 0.769–0.936, and the split-half reliability distribution was 0.695–0.903, thus indicating good reliability and validity.

The Chinese version of the Presence Questionnaire (PQ), used to measure participants’ current sense of presence, was revised by [54]. It comprises 29 items under four dimensions: engagement, sensory reality, adaptation, and interface quality; this questionnaire exhibits good reliability and validity.

The Kirton Adaptation Innovation Inventory (KAI) was used to measure adaptability and innovative cognitive style, including three dimensions of originality, efficiency, and rule/group conforming, with 32 items in total. The lower the score, the more likely the adaptive cognitive style would be. The higher the score, the more likely it is to be in the innovative cognitive style [66].

We used a VR equipment set (virtual glasses set VIVE-P130: two locators and power adapter, streaming box and power adapter, head-mounted equipment and connection cable, and two control handles).

We used the BIOPAC MP160 physiological multichannel instrument EEG module (leads, amplifiers, and several electrode pads).

We used a computer for running virtual reality scenes and connecting VR devices (ALIENWAREX15-1766QW, i7-11800H, 16 GB, 1 TB, and NVIDIA RTX3060).

We used a laptop computer for running AcqKnowledge and connecting to a physiological multichannel instrument (ASUS Zenbook U4700J, i5-1035G1, 16 GB, and 512 GB).

### 2.3. Procedure

A 3 (three experience scenes: urban, garden, and interactive garden) × 2 (two time points: before and after the experience) mixed experimental design was used. A total of 72 participants were randomly divided into three groups (urban group, garden group, interactive garden group), and each participant experienced a VR scene in the experiment. Before the creative ability test, participants were given detailed instructions about the tasks they were expected to complete, and the instructions were consistent. Participants were provided with the same number of practice questions before the formal experiment. Participants first filled out the KAI, and completed the creativity task pre-test. They were then fitted with the physiological equipment for the collection of EEG data from the prefrontal brain area. The skin surfaces of the participants were wiped with alcohol and saline, and then disposable patch electrodes with positive and negative input signals were pasted onto the left and right sides of the forehead; GND electrodes were placed at the temporal bone mastoid behind the ear to measure the prefrontal EEG signal. After participants donned the VR equipment, the physiological multichannel instrument to record their EEG data was turned on; they were instructed to sit still for three minutes to allow EEG baseline recording. All the participants underwent adaptive training before the VR scene experience to ensure that each participant reached a consistent level (able to see the picture clearly, move freely, operate and move with the controller). Participants were randomly assigned to experience only one scene. Thereafter, participants experienced the VR restorative scene for 10 min (Appendix A Figure A2). The experiment was approved by the Ethics Committee of the Department of Psychology, Beijing Forestry University.

Following the experience, the creativity test was completed, and the physiological multichannel instrument EEG module was turned off. The RES and PQ scales were completed, and a short interview was conducted. The reward for completing the experiment was CNY 15. The experimental procedure is shown in Figure 1.

### 2.4. Data Preprocessing and Analysis

To ensure the objectivity of scoring, three graduate students in the Psychology Department scored each participant on the three creativity tests simultaneously, and the average score of the three was used as the final score of the participant. To exclude the influence of cognitive style types on creativity, adaptive participants were excluded, and participants whose data were retained were all innovative in their cognitive style.

IBM SPSS Statistics, version 21.0 (IBM Corp., Armonk, NY, USA), was used for the data analysis. The absolute values of kurtosis and skewness of the data were both less than 1.96, in line with a normal distribution. Repeated measures analysis of variance was used to compare the differences between pre- and post-test. To compare the difference among groups, one-way ANOVA was performed on the pre- and post-test data. In addition, the variation of the creativity tests can be obtained by differentiating the post-test score from the pre-test score. One-way ANOVA was conducted to compare the degree of variations in creativity associated with the VR experience in different scenes.

For EEG, AcqKnowledge 5.0 software (BIOPAC Systems, Inc., Goleta, CA, USA) was used for the digital signal processing of the collected EEG signals. The comb filter was used to set the fundamental frequency to 50 Hz; the IIR recursive filter was used for preliminary filtering, and the EEG low-pass to high-pass filter was set to 1 Hz to 40 Hz to obtain the EEG signals after initial noise reduction. After the EEG signal was filtered using AcqKnowledge 5.0, the artifacts were removed manually in order to prevent interference from events such as eye movements, large movements of the head and body, and sweating. The noises caused by disturbances during the experiment were manually deleted; the signal patterns of each frequency band are shown in Appendix A Figure A1. Further, Matlab 2019a was used for offline denoising and analysis of EEG signals. Then, eye movement artifacts were detected using a sliding window function peak–peak threshold method with MATLAB, and amplitude changes of more than 150 μV were excluded. Drift and other artifacts larger than 100 μV were detected and marked by a cyclic algorithm, then excluded. In order to gather the power spectral density, the Welch method was used to divide the data into 1 s long windows with 50% overlap. The EEG indices of each channel were calculated. Based on Fourier analysis, the fluctuation of surface topography was converted into the intensity spectral distribution of high- and low-frequency topography components in the spatial frequency domain; that is, the power spectral density (PSD) of EEG signals. PSD is divided into δ (0.5–4 Hz), θ (4–8 Hz), α (8–13 Hz), β (13–30 Hz), and γ (30–40 Hz).

The PSD of each frequency band in the scene experience stage was deviated from the PSD in the baseline, and the PSD in the creative task stage was deviated from the PSD in the scene experience stage to obtain two PSD variations. A normality test was performed on the variations, which did not conform to a normal distribution. To compare the differences between groups of changes in the experience of different scenes, ANOVA was conducted and logarithmic transformation was performed on the changes. Following the transformation, the absolute values of kurtosis and skewness of the data were less than 1.96, in line with normal distribution.

Three EEG indicators were calculated according to the PSD of each frequency band [52]: BBR = $\beta_{high}$/$\beta_{low}$, EI = β/(θ+α), and TBR = θ/β represented alertness, engagement, and calmness, respectively. According to the normality test, the data did not conform to the normal distribution. For further analysis, the data were logarithmically converted, and the absolute value of kurtosis and skewness of the converted data was less than 1.96, in line with a normal distribution.

The text analysis software ROST-CM6 (Wuhan University, Wuhan, China) (ROST Content Mining System Version 6.0) was used for analysis. The text of the short interview was divided into words, and word frequency analysis and statistics were conducted using the function of “word frequency analysis” to obtain the high-frequency word list and co-occurrence matrix word list.

Semantic network analysis technology considers the high-frequency words in the text as nodes and the frequency of the common combinations of high-frequency words as the relationship between nodes. Then, it forms a network structure image and turns the scattered concepts into relational knowledge. Visual semantic network analysis images are generated using “Social Network and Semantic Network Analysis” and “NET Draw”. The distance between nodes reflects the strength of the relationship between concepts, and the arrows and lines represent the relationship between the weighted degree of nodes.

Emotion analysis is used to obtain the number and proportion of positive and negative emotions, as well as the number and proportion of entries with different degrees of emotions, and to visually compare the emotional degree of attention elements, relaxation environments, and creative environments in VR environments. Sentiment analysis is based on the combination of a knowledge base (Bosonnlp General Sentiment Dictionary) and corpus (obtained comment data) to conduct sentiment analysis on texts. The Bosonnlp word segmentation corpus is generated by combining news, microblogs, comments, and other sources of data [67]. Compared with other tools, Bosonnlp word segmentation is relatively higher in accuracy. This study adopted the Bosonnlp word segmentation tool to conduct word segmentation based on emotions of collected comment data to obtain the statistical analysis results of positive, neutral, and negative emotions in the text information through quantitative scoring and evaluation of the samples involving emotional expression.

## 3. Results

### 3.1. Restorative Experience of the VR Restorative Environment

The average (standard deviation) of the total score of the RES scale for VR restorative scene was 4.20 (0.74), and the average (standard deviation) of the scores of each dimension was being away 4.47 (1.22), being away, fascination, and compatibility 4.60 (1.16), and extent 2.94 (1.08). The difference test with the mean value of the scale showed that the total score significantly differed from the scale’s mean value. The total score of t(71) = 2.287, *p* = 0.025, the three dimensions of $t_{being\_away}$(71) = 3.277, *p* = 0.002, $t_{fascination\&compatibility}$ (71) = 4.427, *p* < 0.001, $t_{extent}$ (71) = −8.277, and *p* < 0.001. In other words, VR restorative environments precipitated restorative feelings among participants and performed well in the characteristics of being away, fascination, and compatibility, but lacked in terms of the extent of the scene, which should be optimized in future studies.

### 3.2. Presence and the VR Restorative Environment Experience

The descriptive statistical results are shown in Figure 2. The results showed that the total score of presence was *F*(2,69) = 1.748, *p* = 0.182, $\eta^{2}$ = 0.048. Involvement *F*(2,69) = 2.572, *p* = 0.084, $\eta^{2}$ = 0.069; sensory fidelity *F*(2,69) = 4.863, *p* = 0.011, $\eta^{2}$ = 0.124; adaptation *F*(2,69) = 0.804, *p* = 0.452, $\eta^{2}$ = 0.023; interface quality *F*(2,69) = 0.891, *p* = 0.415, $\eta^{2}$ = 0.025.

In addition to sensory fidelity, there was no significant difference between groups in the total score and other dimensions of presence. Further pairings were conducted, the results of which are presented in Appendix Table A1. Sensory fidelity of the interactive experience of restorative environment was significantly higher than that of the experience of the urban and garden scenes without interactive activities.

### 3.3. Influence of the VR Restorative Environment Experience on Creative Performance

#### 3.3.1. AUT

The score of dimensions of the AUT were statistically described, as shown in Figure 3. Repeated measurement ANOVA was conducted for the score of AUT to compare the difference in pre- and post-test. The results showed that there was no significant difference between the total scores on the pre- and post-test, *F*(1,63) = 0.318, *p* = 0.575, $\eta^{2}$ = 0.005. Further, there were no significant differences in fluency, *F*(1,63) = 0.100, *p* = 0.753, $\eta^{2}$ = 0.002; in flexibility, *F*(1,63) = 0.169, *p* = 0.682, $\eta^{2}$ = 0.003; in uniqueness, *F*(1,63) = 0.943, *p* = 0.335, $\eta^{2}$ = 0.015; and in the adherence of the pre- and post-test AUT scores, *F*(1,63) = 0.075, *p* = 0.785, $\eta^{2}$ = 0.001.

The statistical results of the difference between post-test and pre-test scores of AUT are presented in Figure 4.

According to the normality test, all the data conformed to normal distribution. One-way ANOVA was conducted for the change in AUT scores to compare the difference in creativity change through VR experience in the three scenes. The results are shown in Appendix B Table A2. There was no difference between groups in the changes of total score and dimensional scores; that is, there was no significant difference in the degree of changes in AUT between different scenes.

#### 3.3.2. TCT-DP

The pre- and post-test scores of TCT-DP and the scores of each dimension are presented in Figure 5 and Figure 6, respectively. According to repeated measures ANOVA results, there was no significant difference between the total pre- and post-test scores, *F*(1,63) = 2.537, *p* = 0.116, $\eta^{2}$ = 0.039. Repeated measurement ANOVA of other dimensions showed that dimension 10 (Uc_a) exhibited a significant time effect *F*(1,63) = 4.277, *p* = 0.043, $\eta^{2}$ = 0.064, and interaction between time and scene was *F*(1,63) = 4.146, *p* = 0.02, $\eta^{2}$ = 0.116. The interaction between time and scene was significant *F*(1,63) = 5.55, *p* = 0.006, $\eta^{2}$ = 0.15. The post-test of dimension 10 (UC_a) and dimension 13 (UC_d) showed that the garden group had significant differences in the scores of UC_a of the pre- and post-test, $M_{UC\_a0-1}$ = 0.667, *t* (17) = 2.915, *p* = 0.01, 95% CI = [0.184, 1.149], and the pre-test was significantly higher than the post-test. There was a significant difference in UC_a of the post-test between the garden and garden interaction groups, $M_{Uc\_a1\_garden-garden\_interaction}$ = −0.595, *p* = 0.043, 95% CI = [−1.171, −0.019], and the garden interaction group’s score was significantly higher than the garden group. For the urban group, dimension 13 (UC_d) showed a significant difference in the pre- and post-test of the urban group, $M_{UC\_d0-1}$ = −1.1, *t* (19) = −2.979, *p* = 0.008, 95% CI = [−1.8728, −0.3272]; the post-test was significantly higher than the pre-test. For the garden group, $M_{Uc\_d0-1}$ = −2.0556, *t* (17) = −5.916, *p* < 0.001, 95% CI = [−2.7886, −1.3225], the post-test was significantly higher than the pre-test. There was a significant difference between the urban and garden groups on the pre-test of the Uc_d, $M_{Uc\_a1\_urban-garden}$ = 1.206, *p* = 0.004, 95% CI = [0.391, 2.02]; the urban group was significantly higher than the garden group. There was a significant difference in the pre-test of the Uc_d between the garden and garden interaction groups, $M_{Uc\_a1\_garden-gardeninteraction}$ = −1.341, *p* = 0.001, 95% CI = [−2.099, −0.584], and the garden interaction group was significantly higher than the garden group.

Repeated measurement ANOVA of other dimensions showed that dimension 10 (Uc_a) exhibited a significant time effect *F*(1,63) = 4.277, *p* = 0.043, $\eta^{2}$ = 0.064, and interaction between time and scene was *F*(1,63) = 4.146, *p* = 0.02, $\eta^{2}$ = 0.116. The interaction between time and scene was significant *F*(1,63) = 5.55, *p* = 0.006, $\eta^{2}$ = 0.15. The post-test of dimension 10 (UC_a) and dimension 13 (UC_d) showed that the garden group had significant differences in the scores of UC_a of the pre- and post-test, $M_{UC\_a0-1}$ = 0.667, *t* (17) = 2.915, *p* = 0.01, 95% CI = [0.184, 1.149], and the pre-test was significantly higher than the post-test. There was a significant difference in UC_a of the post-test between the garden and garden interaction groups, $M_{Uc\_a1\_garden-garden\_interaction}$ = −0.595, *p* = 0.043, 95% CI = [−1.171, −0.019], and the garden interaction group’s score was significantly higher than the garden group. For the urban group, dimension 13 (UC_d) showed a significant difference in the pre- and post-test of the urban group, $M_{UC\_d0-1}$ = −1.1, *t* (19) = −2.979, *p* = 0.008, 95% CI = [−1.8728, −0.3272]; the post-test was significantly higher than the pre-test. For the garden group, $M_{Uc\_d0-1}$ = −2.0556, *t* (17) = −5.916, *p* < 0.001, 95% CI = [−2.7886, −1.3225], the post-test was significantly higher than the pre-test. There was a significant difference between the urban and garden groups on the pre-test of the Uc_d, $M_{Uc\_a1\_urban-garden}$ = 1.206, *p* = 0.004, 95% CI = [0.391, 2.02]; the urban group was significantly higher than the garden group. There was a significant difference in the pre-test of the Uc_d between the garden and garden interaction groups, $M_{Uc\_a1\_garden-gardeninteraction}$ = −1.341, *p* = 0.001, 95% CI = [−2.099, −0.584], and the garden interaction group was significantly higher than the garden group.

The means (standard deviations) of the change of the TCT-DP score were −1.75 (6.54), 0.333 (5.57), and −0.143 (7.30) for the urban, garden, and garden interaction groups, respectively. The descriptive statistical results of score changes of each dimension are shown in Figure 7.

The ANOVA results of variance of the total and each dimension scores are shown in Appendix B Table A3. The change of dimension 10 (Uc_a) significantly differed among groups, and the change of $M_{Uc\_a\_garden-gardeninteraction}$ = −0.810, *p* = 0.005, 95% CI = [−1.372, −0.247]. There was significant difference in the change of dimension 13 (Uc_d) among groups, $M_{Uc\_a\_urban-garden}$ = −1.233, *p* = 0.007, 95% CI = [−2.115, −0.352]. The change of dimension 13 (Uc_d) of the garden group was significantly higher than that of the urban group. $M_{Uc\_a\_garden-gardeninteraction}$ = 1.262, *p* = 0.003, 95% CI = [0.442, 2.082]; the garden group had a significantly higher change in dimension 13 (Uc_d) than the garden interaction group.

#### 3.3.3. CRAT

The descriptive statistical results of the pre- and post-test scores of CRAT are shown in Figure 8. The results of repeated measures analysis of variance showed that the time effect was significant *F*(1,63) = 48.048, *p* < 0.001, $\eta^{2}$ = 0.433, and the post-test score of CRAT was significantly higher than that of the pre-test, $M_{CRAT1-0}$ = 4.763, 95% CI = [3.39, 6.137]. The interaction between time and scene was significant, *F*(1,63) = 7.08, *p* = 0.002, $\eta^{2}$ = 0.184; the post-test showed that the CRAT score of urban group was significantly higher than that of the pre-test, $M_{CRAT\_urban\_0-1}$ = −4.517, 95% CI = [−6.709, −2.324], *t* (19) = −4.312, *p* < 0.001. The post-test score of CRAT in the garden group was significantly higher than that in the pre-test, $M_{CRAT\_garden\_0-1}$ = −8, 95% CI = [−11.152, −4.848], *t* (17) = −5.355, *p* < 0.001.

The results of one-way ANOVA showed that pre-test, *F*(2,63) = 1.337, *p* = 0.27, indicated no significant difference among groups; the post-test, *F*(2,63) = 44.355, *p* = 0.017, indicated significant differences among groups. The paired comparison is shown in Appendix B Table A5. The CRAT post-test scores of the garden group are significantly higher than those of the urban and garden interaction groups.

According to the the variation of CRAT, the comparison of the degree of variation of creative performance of the three scenes shows that *F*(2,63) = 7.08, *p* = 0.002, $\eta^{2}$ = 0.184. The pairwise comparison results are shown in Appendix B Table A4. Significant differences were found between the garden and garden interaction groups, and the score difference of CRAT in the garden group is significantly higher than that in the garden interaction group.

### 3.4. EEG of VR Restorative Environment Experience and Creativity Task

The statistical results of δ PSD description are shown in Figure 9 and Figure 10. According to the normality test, the data did not conform to normal distribution. For further analysis, the data were logarithmically converted, and the absolute value of kurtosis and skewness of the converted data was less than 1.96, in line with normal distribution.

Repeated measurement ANOVA was conducted for the PSD of different frequency bands; the results, presented in Appendix B Table A6, demonstrate that the EEG time effect of each frequency band is significant, and for the γ wave PSD, the interaction between time and scene is significant. Time variables were further compared in pairs; the results are shown in Appendix B Table A7. The mean PSD of the δ wave was significantly lower at baseline than in the scene experience stage, and significantly lower in the scene experience stage than in the creative task. For the mean PSD of the θ wave, the scene experience and creative task stages were significantly higher than at baseline, and the scene experience stage was significantly higher than the creative task stage. The mean PSD value of the α wave in the scene experience and creative task stages was significantly higher than the baseline, and the scene experience stage was significantly higher than the creative task stage. The mean PSD of the β wave in the scene experience and creative task stages was significantly higher than at baseline, and the scene experience stage was significantly higher than the creative task stage. The mean PSD of the γ wave, scene experience stage, and creative task stage were significantly higher than at baseline, and the scene experience stage was significantly higher than the creative task stage.

Due to the significant interaction of γ wave PSD in time and scene, the paired sample *t*-test of the mean PSD of the γ wave in different scenes was compared in pairs. For the urban group, $t_{urban\_\gamma \_0-1}$(20) = −5.773, *p* < 0.001, $M_{urban\_\gamma \_0-1}$ = −8.647. 95% CI = [−11.772, −5.523]; $t_{urban\_\gamma \_1-2}$(20) = 5.876, *p* < 0.001, $M_{urban\_\gamma \_1-2}$ = 8.339, 95% CI = [5.379, 11.299]; that is, the PSD of the γ wave urban for the PSD of γ wave of garden group, $t_{garden\_\gamma \_0-1}$(19) = 4.506, *p* < 0.001, $M_{garden\_\gamma \_0-1}$ = 6.427, 95% CI = [9.412, 3.441]; $t_{garden\_\gamma \_1-2}$(19) = 4.905, *p* < 0.001, $M_{garden\_\gamma \_1-2}$ = 6.601, 95% CI = [3.784, 9.417]; that is, the PSD of the γ wave of the garden in the scene experience stage was significantly higher than at baseline and in the creative task stage. For the PSD of the garden interaction group, $t_{garden\_interaction\_\gamma \_0-1}$ (27) = 5.001, *p* < 0.001, $M_{garden\_interaction\_\gamma \_0-1}$ = 2.995, 95% CI = [4.223, 1.766]; $t_{garden\_interaction\_\gamma \_1-2}$ = 4.651, *p* < 0.001, $M_{garden\_interaction\_\gamma \_1-2}$ = 2.923, 95% CI = [1.633, 4.212]; that is, the PSD of the γ wave of the garden interaction group in the scene experience stage was significantly higher than at baseline and in the creative task stage. The one-way ANOVA results showed that in the scene experience stage, *F*(2,67) = 6.039, *p* = 0.004, $\eta^{2}$ = 0.153, the difference among groups was significant, and the difference between the urban and garden interaction groups was significant.$M_{urban-garden\_interaction}$ = 5.284, *p* = 0.002, 95% CI = [2.076, 8.491], the PSD of the γ wave of the urban group was significantly higher than that of the garden interaction group. The γ wave PSD of garden and garden interaction groups was significantly different, $M_{garden-garden\_interaction}$ = 3.89, *p* = 0.018, 95% CI = [0.690, 7.105], the γ wave PSD of the garden group was significantly higher than that of the garden interaction group in the scene experience stage. In the creative task stage, *F*(2, 66) = 0.167, *p* = 0.846, $\eta^{2}$ = 0.005. There was no significant difference in γ wave PSD among different scenes.

Figure 11 shows the descriptive statistical results of BBR, EI, and TBR.

Repeated measurement ANOVA was conducted for log-transformed alertness (BBR), participation (EI), and calmness (TBR); the results are presented in Appendix B Table A8. The results demonstrated that the time effect of the alertness and participation indexes was significant, and the time effect edge of the calmness index was significant. The time and scene interaction of the three calculation indexes were significant. The time variables were further compared in pairs; the results are shown in Appendix B Table A9. The indexes of alertness and participation at baseline and in the creativity task were significantly smaller than those in the scene experience stage. The calmness index was significantly higher in the creative task stage and at baseline than in the scene experience stage.

Since the time and scene interaction of the three calculation indexes were significant, a pairwise comparison was conducted between the paired samples of the three calculation indexes in different scenes (Appendix Table A10). For BBR (alertness) of the urban group, the scene experience stage was significantly higher than the baseline, and scene experience stage was significantly higher than the creative task stage. For EI (participation), the scene experience stage was significantly higher than the baseline, and the scene experience stage was significantly higher than the creative task stage. For TBR (calmness), the baseline stage was significantly higher than the scene experience stage, and the creative task stage was significantly higher than the scene experience stage. For BBR (alertness) of the garden group, the scene experience stage was significantly higher than the baseline, and the scene experience stage was significantly higher than the creative task stage. For EI (participation) of the garden group, the scene experience stage was significantly higher than the baseline, and the scene experience stage was significantly higher than the creative task stage. For TBR (calmness) of the garden group, there was no significant difference between the baseline and scene experience stage, and there was no significant difference between the creative task and scene experience stages. For BBR (alertness) of the interactive garden group, there was no significant difference between the scene experience stage and baseline and no significant difference between the scene experience and creative task stages. For EI (participation) of the interactive garden group, the scene experience and baseline stages exhibited no significant difference, and there was no significant difference between the scene experience and creative task stages. For TBR (calmness) of interactive garden group, the baseline was significantly lower than the scene experience stage, and there was no significant difference between the creative task and scene experience stages.

The descriptive statistical results of variations of PSD are shown in Figure 12. To compare the variation of experience of different scenes in PSD of each frequency band, one-way ANOVA was conducted for the change of PSD of each frequency band; the results are shown in Appendix B Table A11. The results demonstrated significant differences between the two stages of the β and γ waves, and significant differences between the two stages of the δ wave. Further pairwise comparison was conducted; the results are shown in Table A12. The change of δ wave PSD in the first stage was significantly greater in the garden interaction group than in the urban group. There was no significant difference in the change of the δ wave PSD in the second stage. The changes of the PSD of β in the first and second stages were significantly higher in the garden interaction than in the urban group. The change of γ of the second stage in the urban and garden groups was significantly higher than that in the garden interaction group.

The difference between the calculated indexes of the scene experience and baseline stages—and that between the creative task and scene experience stages—was calculated; the descriptive statistical results are shown in Figure 13.

To compare the variation of experience in different scenes through the PSD calculation index, one-way ANOVA was conducted for the variation of the PSD calculation index; the results are shown in Appendix B Table A13. There were significant differences in the scenes of changes in alertness and participation. Further pairwise comparison was conducted; the results are shown in Appendix B Table A14. The changes of the first and second stages of alertness in the urban group were significantly greater than those in the garden interaction group. The changes of participation in the first and second stages were significantly higher in the urban group than in the garden interaction group.

### 3.5. Analysis Results of Interview Texts

Following the completion of the experimental task, the experimenter conducted a simple interview with the participants, including the following three questions:What elements of the environment did you focus on (or like) during the experience of the virtual scene?In life, what kind of environment do you think will make you feel relaxed and physically and mentally refreshed?What kind of environment (or scene) do you think smoothly facilitates the emergence of inspiration and creativity? Describe the environment or scene.

A total of 216 valid texts were collected from 72 participants’ answers to the aforementioned three questions. The high-frequency words that focus on scene elements and relaxing environments and stimulate creativity are listed in Appendix B Table A15. According to the word frequency analysis, the elements that participants pay greater attention to in the VR scene experience include the sky, trees, rivers, flowers, sunshine, and other natural scenery elements; additionally, the elements of concern include some manmade elements, such as houses, buildings, rocking chairs, and road signs. In addition to focusing on natural or artificial scenes, participants who experienced interactive activities were more concerned about these in the scene that showed growing vegetables, fishing, flying kites, and feeding birds. For the environments that make participants feel relaxed, the natural scenes with few artificial elements—such as nature, sunshine, quietude, seaside, freshness, grass, and blue sky—were mentioned frequently. In addition to nature, participants further mentioned outdoor activities such as playing with friends, conducting barbecues, and enjoying sports, which helped participants feel relaxed and comfortable in the environment. Some participants mentioned unfrequented scenes, busy markets, or the familiar surroundings of their homes as ways to relax. Regarding the environment that stimulates participants’ creativity, most participants mentioned that a quiet, relaxed, and comfortable natural environment and an environment with no one else is conducive to inspiration, which is consistent with previous studies. Furthermore, some participants stated that environments or scenes that made them feel happy, euphoric, or depressed, such as a dark rainy night, the freedom to let go of their sadness, or a high mood, stimulated creativity. Comparing the high-frequency words focusing on elements, relaxing environment, and creative environment, considerable overlap is evidently found most typically with elements such as sky, trees, rivers, sea, or inaccessible natural scenes.

The semantic network analysis results of concerned elements (Appendix C Figure A6) clearly show that the scene elements that participants are mainly concerned about belong to two themes: (1) natural elements, including those diverting from the sky and trees; (2) artificial elements, including those emanating from roads and houses. Analysis results of the relaxed environment of the semantic network are presented in Appendix C Figure A7. The elements and character descriptions of the relaxing environment are mainly divided into two categories, the first including animals and plants, sea, water, sky, sun, and other natural elements of indoor or outdoor environments. The roof and outdoor artificial environments and the playground are in the second category. Environment analysis results, based on the semantic network of creativity, are shown in Appendix C Figure A8. The center sends out creative environment elements and features descriptions and relaxing environments that have significant similarities to comfortable and peaceful natural environments. Colorful visual stimulation eliminates noise disturbance, such as the unmanned environment, and is more likely to stimulate individuals’ creativity.

The comparative analysis results of positive and negative emotions are shown in Appendix C Figure A3. The proportion of positive emotion in the relaxed environment was the highest, followed by that in the creative environment, while that in the VR environment was the lowest. The proportion of neutral emotion in the VR environment was the highest, which was significantly higher than that in the relaxed and creative environments. The emotional types of the three environments concentrated on positive and neutral emotions and showed minimal negative emotions. In the three environments, the creative environment exhibited the highest proportion of negative emotions.

The degree of positive and negative emotions in the three environments were compared; the results of comparative analysis between positive emotion and negative emotion intensity are shown in Appendix C Figure A4 and Figure A5. Whether in the VR, relaxation, or creative environments, positive and negative emotions exhibit similar performance in emotional intensity, and they predominantly exhibit a prevalent intensity.

The proportion of positive emotions in the relaxation and creative environments described by participants was significantly higher than that in the current VR environment, indicating that there is room for optimization in the current VR scene. The negative emotions in the creative environment described by participants were significantly higher than those in the VR and relaxed environments; the proportion of general and moderate negative emotions in the creative environment was the highest. Neither the current VR environment nor the relaxation environment described by the participants exhibited moderate or high negative emotions; on the contrary, the creative environment described by the participants exhibited moderate negative emotions.

## 4. Discussion

### 4.1. Restorative Experience of the VR Restorative Environment

According to the RES score results, the VR scene that we used conforms to the characteristics of a restorative environment and is capable of precipitating restorative feelings among the participants. Previous laboratory studies have shown that natural settings form a better restorative environment than urban settings [68]. Urban environments with natural elements, architectural elements with cultural and leisure functions, and differences between streets and residential areas are also restorative environment settings [69]. Environments that interact with places associated with leisure activities, especially those where social activities can occur, exhibit a high restorative potential. Since social activities are more accessible in urban environments, with certain conditions, they may also act as effective restorative environments [70]. From an aesthetic perspective, an urban environment that has natural elements and is well-planned and well-designed is more likely to not only be preferred by people but also provide restorative effects. Meanwhile, research shows that the impact of environmental stimulus on emotional changes is based on the mechanism of aesthetic preference: whether it is an urban or natural environment, the environment type in line with high aesthetic preference invariably has the greatest impact on emotional changes [71]. Attractive places are considered more aesthetic and, therefore, more restorative, and the sense of belonging to urban settings will affect people’s subjective feelings of restoration [69,72]. The presence of people, and types and heights of buildings are also important factors affecting people’s feelings on urban environment restoration [73]. The urban scene provided in the experiment exhibits a wide variety of buildings with no human presence, and a more empty and quiet feeling than that of a real environment, which is different from the pressure environment that participants are familiar with and that is in line with their aesthetic preferences. Interactive garden scenes provide interactive activities that infuse a sense of interest and entertainment.

The positive correlation between the aesthetic preference and restorative effect has also been verified; that is, setting more trees, brightly colored flowers, and providing clean water in the environment enhances the restorative effect as well as aesthetic preference [74]. Current research on the relationship between restoration and aesthetics has gradually deepened and found that the assessment of the effect of environment restoration can better predict the degree of preference, which applies to natural and urban types of scenes; this relationship can be moderated by cognitive (perceived restoration) or emotional (positive emotion) processes [75]. The literature further confirms the inseparable relationship between the two. Therefore, discussing the restorative feeling and effect from the perspective of aesthetic preference may be beneficial. Regarding the dimensions of restorative characteristics such as attraction and compatibility, an environment with high aesthetic preferences provides individuals with opportunities to reflect on important experiences and issues, to obtain a deeper restorative effect through aesthetics. Urban green spaces are now more designed and controlled and have certain activity service functions; different plant spaces promote different behaviors and activities. Consequently, experiencing such an environment facilitates high attraction and compatibility and a desirable feeling of recovery. Additionally, studies have shown that the role of attraction may be highly relevant in the restoration process of architecture and historical environments. Attraction can be enhanced by changing the existing characteristics of public space in a more artistic direction and by properly maintaining high-value artistic and historical urban areas, suggesting that the restorative potential of environments that combine nature and art is noteworthy [76].

The ability to perceive the restorative potential of an environment further depends on the level of affinity with natural elements. A study of outdoor environments in campuses showed that having a higher natural perception can increase the perception of environmental restoration [77]. Studies focusing on urban green spaces with different degrees of naturalness also show that the perception of environmental restoration increases with the increase of the naturalness of the environment [78]. Furthermore, perceived restoration may depend on an individual’s connection to nature and this relationship may also vary with the biophilic quality of the environment, i.e., the functional and aesthetic value of the natural environment [79]. Different ages have different perceptions of the restorative capacity of urban and natural settings. A study comparing age-based perceptions of restoration found that children preferred urban environments, while adults preferred nature [80]. Regarding personality traits, compared with people with low neuroticism scores, people with high neuroticism scores may exhibit greater benefits after exposure to urban environments [81]. Furthermore, studies have shown that urban nature can effectively replace wild nature when planning restorative environments [82], indicating that the key aspect that affects the perception of the level of environmental restoration may not be a change in in the type of environment (e.g., streets, buildings), but rather the characteristics of specific (objective or subjective) elements of the environment (e.g., water quality, greenery quality, biodiversity). Therefore, the discussion on the influence of environmental restoration can shift the focus from comparing urban and natural environments, real and simulated environments, and the effectiveness of measurement tools to distinguishing the elemental or individual characteristics of varied environments [83].

### 4.2. Presence and the VR Restorative Environment Experience

According to the results of presence in VR scene experiences, the sensory reality of the garden interaction group was significantly higher than that of the urban and garden groups. The interaction and reaction between human and environment in the VR experience may be mediated by the sense of presence. Compared with the 2D virtual experience, the 3D VR experience has advantages in improving the sense of presence, and an interactive VR is associated with more significant improvement in positive than negative emotion. This may be mediated by an increased sense of presence and connectedness to nature [39]. It is generally believed that being able to interact with the environment rather than passively observing it makes people feel more present in the environment [84]. Activities such as bird feeding, watering flowers, kite flying, and fishing in the interactive garden group made participants use their bodies in natural and dynamic ways, such as bending, squatting, reaching out to pick up objects, walking within a certain range, and overlooking, which may produce a stronger sense of presence [85]. Cognition is embodied, and physical activities enhance an individual’s environmental perception. Therefore, participants experienced the strongest sensory fidelity in VR scenes with interactive activities.

Moreover, the sense of presence is also related to the vividness and sense of control over mental imagery; that is, the ability to generate vivid visual images is positively correlated with the sense of presence in VR [86]. The sense of presence is higher when mental representation is generated spontaneously rather than merely perceiving the virtual environment presented by VR. Virtual experience is similar to “realistic” mental imagery, which has also been proven in experimental research on the sense of presence of virtual architectural heritage. Research shows that people’s feeling of being in a place is usually unique and personal, which is influenced by their memory and experience of the local environment through visual perception [87]. Therefore, the sense of presence in VR is related to individual differences in mental representations, especially the ability to create vivid and clear mental representations. From this, the results should show that the urban group exhibits a higher sense of presence because people are more familiar with urban scenes in daily life. However, the sense of presence in the garden interaction group is higher than that of the urban group, which may be because the activities in the interaction group improve the sense of immersion through embodied movement. Compared with simply simulating walking and watching in urban or garden groups, the activities in the interaction group are more conducive to generating vivid mental representation and stronger sensory fidelity, both of which improve the sense of presence.

Moreover, immersion technology moderately impacts the sense of presence; compared with improving the functions of other immersive systems (including the quality of visual and auditory content), enhancing the level of user tracking, the use of stereoscopic vision, and a wider visual display field are significantly more influential [88]. Comparing the differences in the impact of the two control modes (glove vs. controller) and the two grasp visualizations (tracking hand vs. disappearing hand) on the user experience shows that presence is significantly increased when hand tracking (glove) is used as the input mode [89], which suggests that the interactive input mode of the controller used in this study can be further improved in the future to obtain a higher sense of presence.

### 4.3. Influence of the VR Restorative Environment Experience on Creative Performance

According to previous research findings, nature is generally considered more restorative than the urban environment, but its benefit to people’s cognitive resources may not be much greater than that of the urban environment. Measuring restorative perception without directly measuring the impact of cognitive function may lead to misleading results [90]. We used three methods to directly measure the impact of VR restorative environments on individual creative performance, and compared the differences in creativity performance before and after the experience.

#### 4.3.1. AUT

The results showed that there was no significant difference in the total score and dimension score of the pre-and post-test of AUT, indicating that VR scenes in this study did not significantly impact participants’ divergent thinking task performance. However, we cannot draw a conclusion that the restorative environment has no beneficial effect on divergent thinking. According to the perception–action perspective framework, creativity depends on the initial stimuli in the situation and on the extent to which the individual explores, perceives, and exploits the (uncommon) action possibilities that the situation affords. This external visual perception process can also occur on the basis of imagination or mental representation, guided by internally directed attention [91,92,93]. According to previous studies on the effects of stimulation on creativity in the immediate context, the presence of different real objects in the immediate context will affect children’s original creative performance [92]. Studies on perceived environmental cues’ impact on creativity have also shown that individuals respond with greater originality and complexity to creativity tasks in the virtual environment conducive to creativity, with no differences in fluency and flexibility [94]. These results prove that stimulus cues in the context significantly impact divergent thinking. However, considering this study’s scene settings, this result may be due to the lack of stimulus cues related to the AUT task in the restorative scene; thus, the virtual restorative scene experience did not produce a sensitive difference in the performance of individual divergent thinking tasks.

Additionally, studies have shown that a lack of focus is beneficial to problem-solving [95,96], which is also consistent with the observation that ADHD patients perform with high levels of fluency, flexibility, and originality in the AUT, as well as achieving high creativity scores on the Creative Achievement Questionnaire [97]. The results of studies on the influence of the physical state on divergent thinking show that the more control resources physical activity consumes, the easier it is for divergent thinking to develop, which is possibly because less top-down control leads to more ideas [98]. Therefore, based on ART, natural scenes exhibit a restorative effect on directed attention; virtual restorative experiences restore cognitive resources; and the restoration of directed attention may generate more top-down cognitive control, which inhibits the development of divergent thinking, demonstrating that the virtual restorative environment experience does not significantly impact AUT test results. Moreover, compared with studies that also used AUT to measure creativity, there is no significant difference in results before and after experience, which may be due to the fact that the effectiveness of environmental manipulation has not been verified and guaranteed. It may increase the effectiveness of environmental manipulation, resulting in more sensitive research results if participants continue to complete tasks in the manipulated environment, and are explicitly asked their views on the environment to increase their attention to it [99]. This also suggests that we should refine the experimental design in future studies to improve the effectiveness of VR manipulation of scene factors, to ensure more sensitive creativity test results. VR scenes in this study did not significantly impact participants’ divergent thinking task performance; it may be that a lack of focus is beneficial to problem-solving, and the restoration of restorative-environment-directed attention may inhibit the development of divergent thinking, so the impact of the restorative environment on divergent thinking remains to be tested. In addition, the method may improve the sensitivity of creativity test results, which is to refine the experimental design, ensure the consistency of the participants’ attention to the environment, improve the effectiveness of VR manipulation of scene factors, and eliminate unnecessary errors.

From the AUT results, we found no differences in each dimension before and after the environment experiences. Findings in research exploring how to introduce creativity through education found that all dimensions of students’ divergent thinking have changed, and that a close correlation exists between each dimension; the significant increase in both originality and fluency can be explained by the fact that an original idea is more likely to be found when the number of ideas generated increases [100], and is more likely to produce one that fits into another concept category [101]. To some extent, the close relationship between the dimensions of divergent thinking explain this study’s result that no significant difference was found in the AUT dimension between before and after the scene experience.

#### 4.3.2. TCT-DP

Through the TCT-DP, a comprehensive representation of participant creativity, we found significant interactions between time and scene Uc_a (dimension 10: any manipulation of the material), and Uc_d (dimension 13: unconventional use of given fragments). This could be attributed to the fact that personal characteristics (e.g., age, sex, education level, or personality) may significantly contribute to determining cognitive recovery. Both urban and garden experiences enhance the picture presentation in the unconventionality aspect, indicating that the restorative environment experience through VR may be beneficial to creative expression. Additionally, the score of participants who experienced the garden scene in dimension 13 is significantly higher than that of the score of interactive garden, which may be related to the affordance of the VR environment. A study exploring the relationship between aesthetics, affordance, and children’s creativity reported that restricting the behavioral possibilities provided by the environment limits creative thinking and behavior. Compared with traditional swing and seesaw facilities, which stimulatechildren’s actions, abstract, attractive, and aesthetic environments attract children to cultivate their ability to independently explore the affordance of the environment and stimulate creativity [102]. Some interactive activities in the interaction group only allow specific and repeated behaviors, resulting in low playability of the activities and failing to stimulate participants’ interest in self-exploration and creativity, which may result in no difference in the change of the novelty dimension in creative expression.

In conclusion, our results not only demonstrate that nature enhances creativity but also provide strong evidence for the stimulation of creativity by VR simulations of nature. VR can mimic nature. A highly immersive virtual environment can stimulate visual interest, excitement, and freshness, and achieve the effect of entertainment, stimulating intelligence and cognitive ability, so as to promote innovative behavior that helps improve professional knowledge and innovative skills, stimulate motivation, and enhance creativity. When people participate in a suitable environment, even if the environment is not real, such as a simulated natural environment, it may influence their behavior and perception and may stimulate their creativity in a certain aspect [59]. The results prove that the VR restorative environmental experience plays an evident role in improving the ability of convergent thinking. However, it is still unclear whether the VR restorative environmental experience, and different types of environmental and interaction modes, improve other aspects of creativity, which can be used as the direction of further research.

#### 4.3.3. CRAT

The post-test scores of CRAT were significantly improved after the VR restorative environment experience. According to previous studies, the three basic features of VR—immersion, interactivity, and imagination—are closely related to the performance of creativity [58]. People are more likely to show creativity during interactive processes or highly immersive interactions. This suggests that VR technology may be an effective tool to improve creative performance, especially the insight test performance, which measures the quality of convergent creativity. Studies have shown that walking into nature or looking at natural images can improve directed attention, supporting ART [29], which helps participants complete creative tasks related to convergent thinking. From the perspective of VR environment experiences, viewing natural images improves cognitive ability, which indicates that nature can potentially promote creative performance, arouse visual and exploration interests, stimulate intelligence and cognitive ability, and promote creative behavior [29,103].

The significantly higher CRAT scores in the garden group compared to the urban group may be related to the physical characteristics of the environment. The low-level visual features of the environment will impact people’s cognitive thinking. The more natural the environment is, the more likely a natural theme of thought will be generated. Further, the environment with non-straight edges can also inspire symbolic and reflective thinking about spirituality and life, and produce positive and calm emotions [104,105]. Therefore, a more natural environment with more non-straight edges in the garden group may be more likely to generate fluent associative cognitive thinking and improve creative performance.

In the comparison of the degree of change among the three groups, the change of CRAT in the garden group was significantly higher than that in the garden interaction group. This may be related to the participants’ adaptation to the VR environment and the consumption of cognitive resources by the interactive activities. Participants who are familiar with VR equipment may consume different cognitive resources from those who are exposed to the VR environment for the first time in the same restorative VR environment experience; the degree of change of creativity performance is also different. In studies exploring the effects of sports or musical stimuli on verbal creative performance in laboratory settings, no difference in effects on creative thinking performance were observed, suggesting that certain stimuli may affect the shift from cognition to narrow attention and cause attentional contraction that may limit creative expression [106]. Therefore, in this study, when subjects started the creativity test immediately after interacting in the VR environment, they may have been in a state of perceptually narrow attention, which hinders creativity performance, thus resulting in lower interactive group scores than the other groups.

### 4.4. EEG of VR Restorative Environment Experience and Creativity Task

In this study, EEG data were recorded at baseline, virtual reality healing environment experience, and creativity test states to compare the differences in EEG activity between the three states. During virtual reality experiences, brain activity can be continuously recorded to reveal ongoing EEG activity in specific frequency bands, such as synchronization and desynchronization of α and β [107]. The examination of emotional and cognitive processes through these EEG oscillations has found a correlation between cognitive needs and changes in these oscillations [108]. It is mainly involved in θ (3–7 Hz), low α (8–10 Hz), and β states (13–30 Hz) [109,110]. From the perspective of time variation, the average PSD of the δ wave increases gradually with time from the baseline state to the scene experience and then to the creative task. The mean power spectral densities of the θ, α, β, and γ waves were significantly increased in the scene experience and creativity task stages, and the mean power spectral densities of the scene experience were the highest among the three stages. The results showed a time–scene interaction of mean γ wave PSD, with the highest mean γ wave PSD during the experience of the city, garden, and garden interaction scenes (higher than the baseline and creativity task stages). From the perspective of different scene experience types, the difference of the average PSD of the γ wave in the process of different scene experiences was found. The urban group had the highest average PSD, followed by the garden group; the garden interaction group had the lowest average PSD. Therefore, our study compared the PSD of different brain waves in different stages, which intuitively reflected the changes of different brain waves in different stages.

Additionally, three brain function indicators, alertness, engagement, and calmness, were selected. Through EEG feedback, it was found that different experimental scenarios had different effects on the prefrontal lobe, and three different stages of prefrontal electrical changes were observed during the baseline, scene experience, and creativity test stages. In the stage of the VR restorative environmental experience, participants exhibited the highest level of prefrontal alertness and engagement, and the lowest level of calmness. However, EEG indicators only reflect the activity of the prefrontal lobe at different stages, and the mechanism of the effect of the VR restorative environment on creativity cannot be clarified.

### 4.5. Analysis Results of Interview Texts

According to the text analysis results, participants pay greater attention to natural elements and some artificial elements when experiencing VR scenes, and interactive activities attract a great degree of their attention. Natural scenes and outdoor activities help participants feel relaxed and comfortable in the environment, which is consistent with previous studies showing that urban green spaces with more sensory dimensions of tranquility and nature and fewer cultural and social dimensions are considered the most restorative places for people under stress [111]. Additionally, participants mentioned that the deserted scene, busy market, or familiar surroundings at home make them feel relaxed. Quiet, relaxed, and comfortable environments in nature without people are conducive to inspiration and are more likely to promote creativity. Solitude can help people spark new ideas, gain insight into their basic values, solve problems more effectively, and feel calm and relaxed; the environment that allows them to be emotionally engaged may stimulate creativity. Previous studies have shown that creative workers who escape the noisy environment for attention recovery and spend time in a quiet space may produce enhanced creative performance [112]. The lack of space for solitude, reflection, and tranquility may hinder creative thinking [113].

Semantic network analysis focusing on elements found that creative environment elements and feature description exhibit significant similarities with relaxed environment characteristics. A natural environment that makes people feel comfortable and peaceful, visual stimulation with bright colors, and a deserted environment that eliminates noisy interference are more likely to stimulate creativity. According to emotion analysis, a relaxing environment provides people a positive emotional connection. The emotional types of VR, relaxing and creative environments, focus on positive and neutral emotions of moderate intensity. However, the negative emotions in the creative environment described by participants were significantly higher than those in the VR and relaxation environment, indicating that the environment that can stimulate creativity may be associated with some negative emotions.

### 4.6. Innovation and Significance

Current studies on restorative environments focus on physical and mental health aspects such as individual mood improvement and stress relief, with a few studies beginning to focus on such environments’ impact on higher cognitive processes. More practical evidence is needed to apply the effect of restorative environmental experience on creativity in practice. At the same time, previous studies on restorative environments are primarily based on the real setting, and few studies explore the effect of the restorative environment in the VR setting. Therefore, from the perspective of the practical application of restorative environment and the experiencer, this study combines advanced technology to bring the restorative environment into VR, and uses EEG as an objective index to explore the impact on creativity.

Combined with the basic theory of restorative environments, the influence of the difference of the restorative environment experience and sense of presence in VR on creativity is explored. In terms of research paradigm, the pre- and post-test design commonly used in restorative environment research was used to compare the differences in participants’ performances before and after environment experience. The use of VR technology combined with classic psychological experimental paradigm and scale evaluation is persuasive. At the same time, physiological instruments were used to obtain EEG data during the experience process, and the physiological information was used to support the scale data to make the study more rigorous.

Theoretically, this study is an exploration and extension of the research on restorative environments, which contributes to the understanding of the internal mechanism of restorative environments on individuals, and also provides new evidence supplementary to the research on the restorative environment experience’s impact on creativity. Meanwhile, from the perspective of practice, this study provides a theoretical basis for the practical application of restorative environments. The advantages of the combination of restorative environment and virtual reality technology overcome the limitations of time and space to find a novel feasible way for urban teenagers and college students to stimulate and improve their creativity, making it possible for them to benefit from the experience of restorative environments, which have practical value as well as significance.

### 4.7. Limitations

First, to compare the effects of visual stimuli only, we abandoned auditory and olfactory sensory systems, which are important for the complete perception of restoration. Some studies have proven that nature is still superior to city as a restorative space when comparing natural sound and urban recordings only. The acoustic environment and the need for recovery and others’ presence have significant influence on the restorative effect of the environment [114]. The natural sounds of birds, wind, and water enhance the positive perception of natural environments through visual representation, and sound improves emotional and cognitive performance subjectively and objectively after stress or fatigue [115]. Therefore, future research must focus on real environment simulation and explore the impact of multisensory recovery environment experience on people.

Second, individual differences, such as individual experience, fluid intelligence, personality (such as openness), and sensitivity to environmental influences (field independence–field dependence), all have important influences on creativity performance. Future research should not only explore the influence of scene types or environmental elements on individual creativity from the perspective of the external environment but also explore whether groups with individual differences, such as groups with different personality traits, exhibit differences in environmental perception and creative performance in the same virtual restorative environment.

Third, we lacked control over the difficulty of the interactive activities. Since numerous participants had not been exposed to VR equipment previously, it may have been difficult to complete the interaction, which consumes cognitive resources and affects subsequent creative task performance. The difficulty of interactive activities should be properly controlled in virtual scene design. While enhancing physical participation and experiencing enjoyment, the influence of activity difficulty on emotion and cognition should be considered to discover the interactive activities suitable for most experiencers.

Fourth, we did not consider the possible effects of the preferences and familiarity on environment restoration. This study only judged whether the environment had the restoration from the perspective of the type of environment. However, the participants’ understanding, interpretation, and perception of the scene are also the keys to affect the perception of the restorative properties of the environment. It is mentioned that the restorative perception not only changed by the type of environment, but also depended on the individual’s aesthetic preference and natural perception. There are significant individual differences in the preferences of environment types and elements. The environment that people prefer will make them feel physically and mentally happy, which itself has the effect of restoration. For example, for participants who prefer a modern urban environment, the urban environment provided may be a good material for their recovery, which is better than a garden environment with more natural elements. Previous studies have found that only the environment of individual attachment has the effect of attention recovery and emotional priming, and adolescents’ local attachment level affects the effect of the natural environment on attention recovery. Researchers have argued that a place with positive memories and emotions may be a restorative environment for mental health [116]. Future studies must focus on the influence of physical environmental characteristics, such as the contrast between nature and city and green-looking ratio on individual psychological state or function. Future research must further explore the restorative effect of place attachment and aesthetic preferences for an environment on the body and mind from the perspective of the relationship between people and environment, eye-tracking technology or subjective evaluation can be used in research, and further discussion can be carried out in combination with individual aesthetic preferences and natural perception. Place attachment can provide multiple psychological benefits [117], such as emotional and cognitive recovery, and help people overcome daily stress [38,118,119,120]. It may be argued that for individuals, no matter what type of environment it is, the environment they prefer and attach to is the restorative environment [120].

## 5. Conclusions

The results of this study show that VR restorative environment experience, as applied in this study:

(1) Improves individual creativity, especially the creative quality of cohesion.

(2) Enables participants to experience a desirable sense of presence. Compared with the restorative scene experience without interactive activities, the addition of interactive activities improves the individual sensory fidelity to a greater extent.

(3) Does not differ between interactive versus non-interactive in improving individual creative performance. Interaction with certain difficulty will increase cognitive load, thus hampering creative performance. Garden scenes that can be explored freely and that have no interaction can better promote individual creativity.

(4) Proves that participants paid more attention to natural elements, and the restorative environment they described was highly similar to the environment that they believed could foster creativity.

## Figures and Tables

**Figure 1 ijerph-19-12083-f001:**
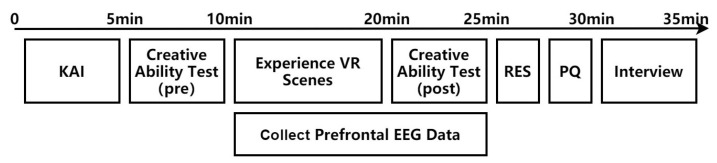
Procedure.

**Figure 2 ijerph-19-12083-f002:**
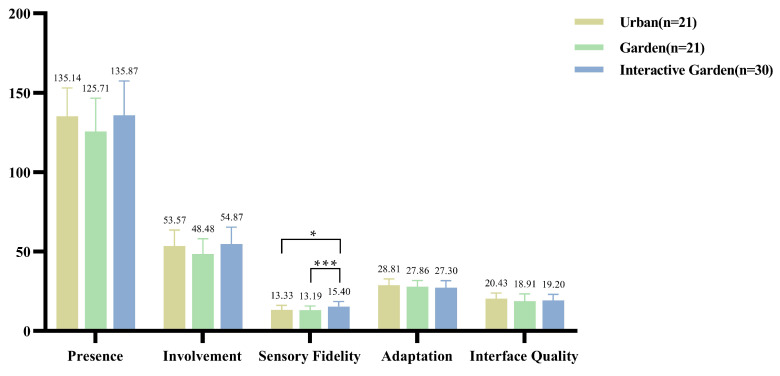
Descriptive statistical results of the total score and each dimension score of the Presence Questionnaire. Note: * represents *p* < 0.05, *** represents *p* < 0.001.

**Figure 3 ijerph-19-12083-f003:**
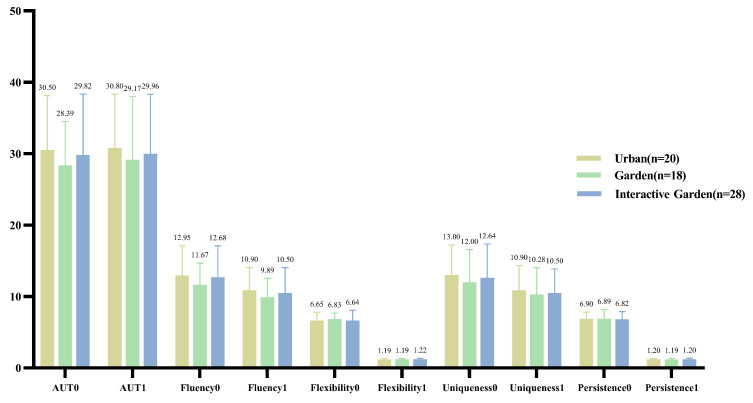
Descriptive statistical results of the scores of AUT.

**Figure 4 ijerph-19-12083-f004:**
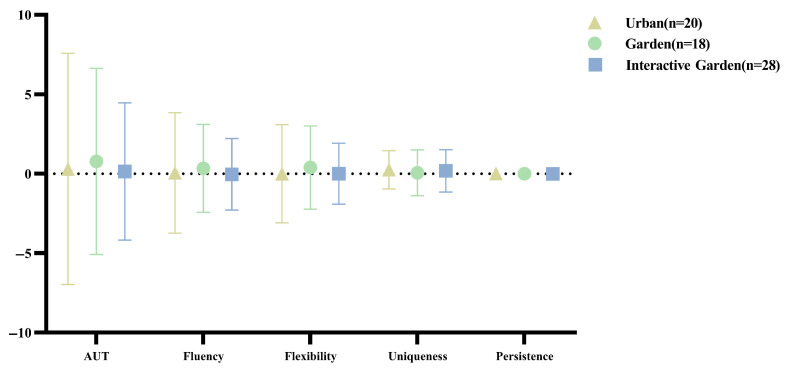
Descriptive statistical results of the scores of AUT.

**Figure 5 ijerph-19-12083-f005:**
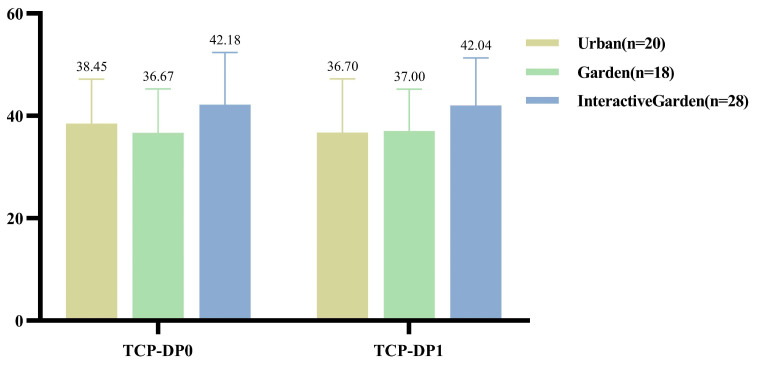
Descriptive statistical results of the total score of TCT-DP.

**Figure 6 ijerph-19-12083-f006:**
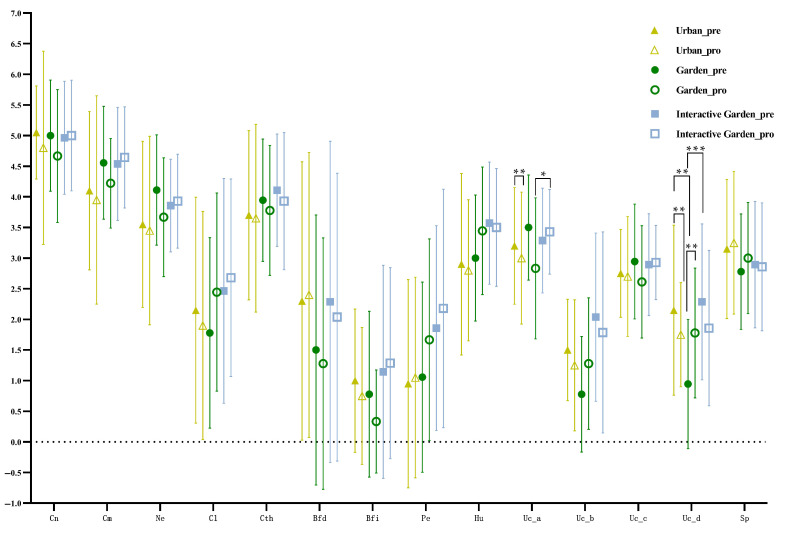
Descriptive statistical results of TCT-DP dimension score of the pre- and post-test. Note: * represents *p* < 0.05, ** represents *p* < 0.01, and *** represents *p* < 0.001. 1: Continuations (Cn); 2: completion (Cm); 3: new elements (Ne); 4: connections made with a line (Cl); 5: connections made to produce a theme (Cth); 6: boundary breaking that is fragment-dependent (Bfd); 7: boundary breaking that is fragment-independent (Bfi); 8: perspective (Pe); 9: humor and affectivity (Hu); 10: unconventionality, a (Uc_a); 11: unconventionality, b (Uc_b); 12: unconventionality, c (Uc_c); 13: unconventionality, d (Uc_d); 14: speed (Sp).

**Figure 7 ijerph-19-12083-f007:**
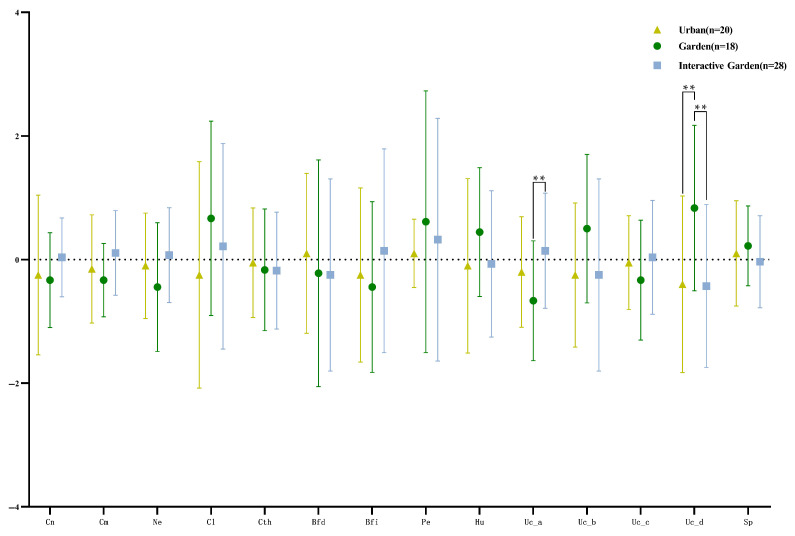
Descriptive statistical results of the TCT-DP score changes for the three scenes in each dimension. Note: ** represents *p* < 0.01. 1: Continuations (Cn); 2: completion (Cm); 3: new elements (Ne); 4: connections made with a line (Cl); 5: connections made to produce a theme (Cth); 6: boundary breaking that is fragment-dependent (Bfd); 7: boundary breaking that is fragment-independent (Bfi); 8: perspective (Pe); 9: humor and affectivity (Hu); 10: unconventionality, a (Uc_a); 11: unconventionality, b (Uc_b); 12: unconventionality, c (Uc_c); 13: unconventionality, d (Uc_d); 14: speed (Sp).

**Figure 8 ijerph-19-12083-f008:**
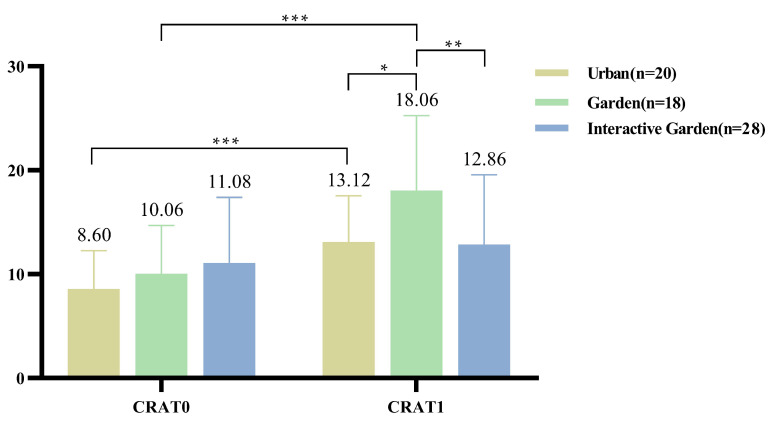
Descriptive statistical results of the total score of CRAT. Note: * represents *p* < 0.05, ** represents *p* < 0.01, and *** represents *p* < 0.001.

**Figure 9 ijerph-19-12083-f009:**
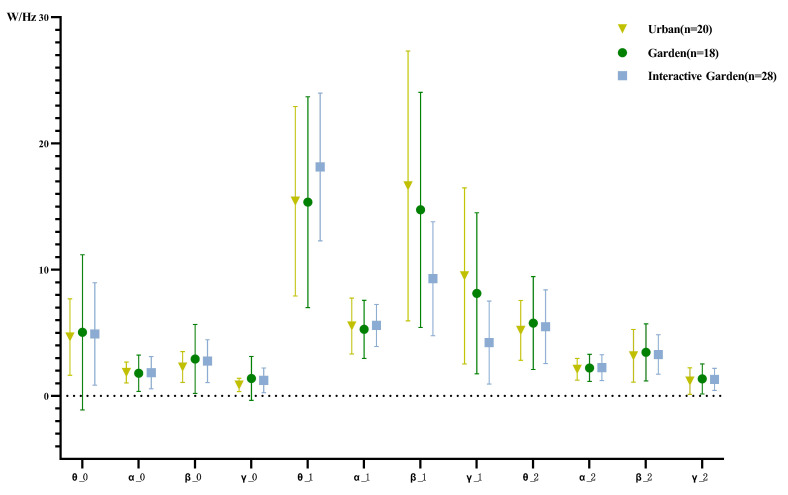
Descriptive statistical results 1 of PSD in different frequency bands.

**Figure 10 ijerph-19-12083-f010:**
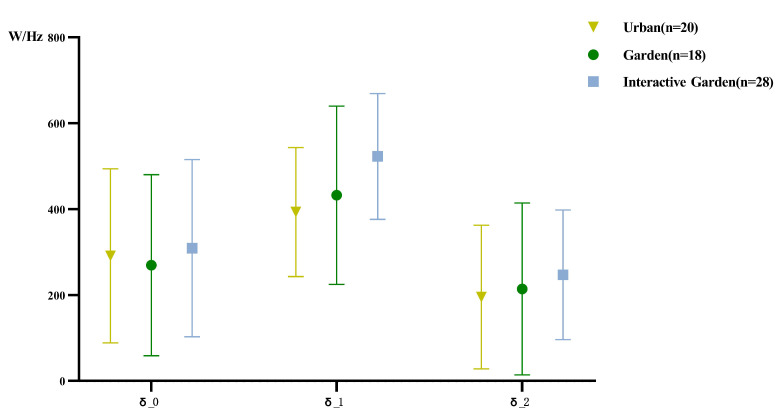
Descriptive statistical results 2 of PSD in different frequency bands.

**Figure 11 ijerph-19-12083-f011:**
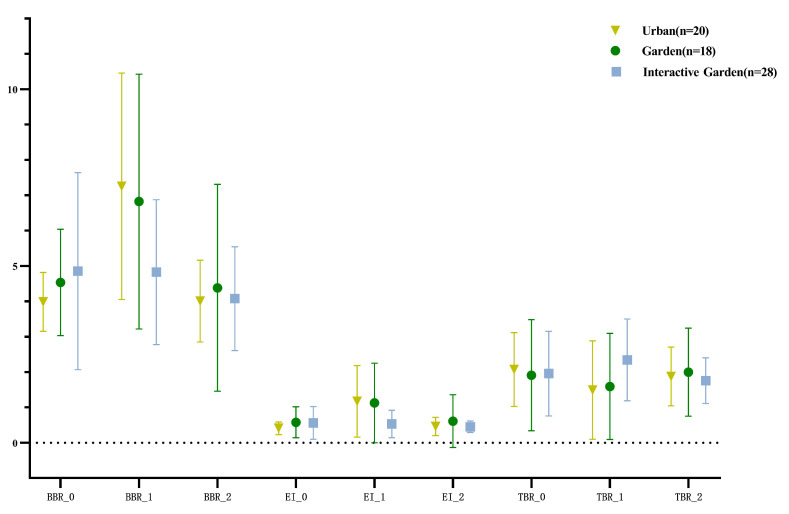
Descriptive statistical results of PSD calculation indicators.

**Figure 12 ijerph-19-12083-f012:**
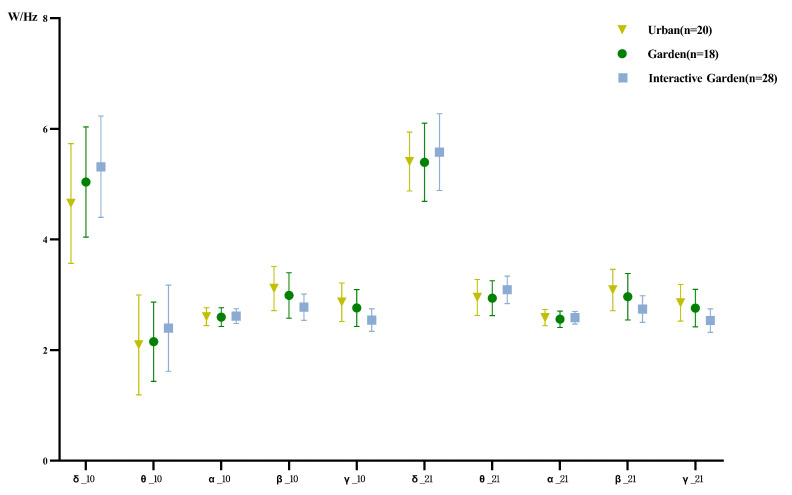
Descriptive statistical results of the change of PSD in different frequency bands.

**Figure 13 ijerph-19-12083-f013:**
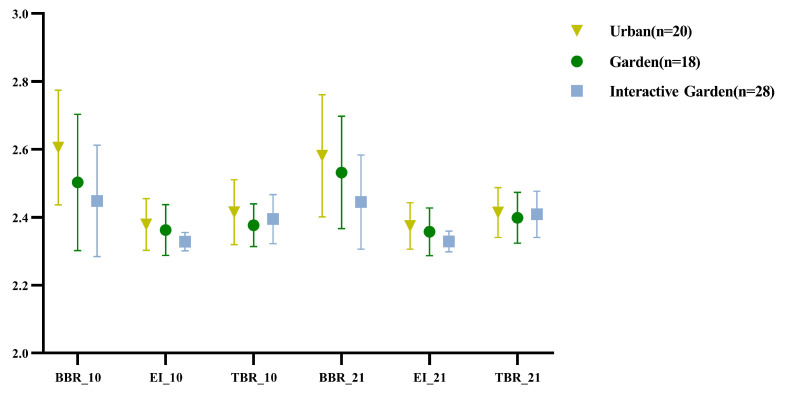
Descriptive statistical results of the change of PSD in different frequency bands.

## Data Availability

The raw data for this article were provided by authors.

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
