# Peer review of "The Effect of Virtual-Reality-Based Restorative Environments on Creativity"

_ijerph, 2022, doi:10.3390/ijerph191912083_

Round 1

Reviewer 1 Report

Congratulations on the study. Very interesting.

I would like to make some recommendations.

You have to follow the CONSORT methodology for your study.

You say that you have done 72 full-time undergraduate and postgraduate students, how did you arrive at this final number, what is the average age, weight, etc...? MORE DATA OF THE PARTICIPANTS IS MISSING

WHAT TYPE OF STUDY IS IT?

In the introduction you talk about hypotheses, you would have to take it to study objectives and not to divide therefore the conclusion.

Author Response

Please check the PDF attachment!

Reviewer 2 Report

The article gives account of a very extensive study that collected many datasets, including EEG measurements, by offering 72 participants (in groups?) different assignments (creative thinking, VR, and answering questions) according to specific classification strategies (like AUT, TCP_DP, CRAT, RES, and PQ) which each consists of many variables (dimensions). The data have been mostly analysed by ANOVA in relation to the normality test (and data conversion if needed). Comparisons before and after VR experience, as well as relationships between variables, were studied. Based on all results four assumptions (named hypotheses) on the influence of the experience of the Virtual restorative environment on individual creativity were discussed and concluded. 
As said, quite a lot of data has been collected and studied, which is impressive. However, the article needs a serious update before it could be published. The update must consider: shortening the total text (overcome repetitive text; limit information, there is too much <like figures**, tables, values mentioned in the text>) which deviates the reader from the main messages), articulating the content (skip text parts, use annexes), reformatting text (avoid listings in the text; make use of short, active sentences; stick to the same terminology), elaborating the procedure of assignments (what test groups?in what order did they perform the assignments?); What do the scenes look like and why?. This information is crucial to help understand what has been measured and why.

Besides, there are many more questions in detail which I partly enclose in the attached PDF.  

** the figures that show the changes of AUT, CRAT, TCT-DP and PSD dimensions (variables) are great

Author Response

Please check the PDF attachment!
